# Molecular Epidemiology of Carbapenemases in *Enterobacteriales* from Humans, Animals, Food and the Environment

**DOI:** 10.3390/antibiotics9100693

**Published:** 2020-10-13

**Authors:** Gurleen Taggar, Muhammad Attiq Rheman, Patrick Boerlin, Moussa Sory Diarra

**Affiliations:** 1Guelph Research and Development Center, Agriculture and Agri-Food Canada (AAFC), 93, Stone Road West, Guelph, ON N1G 5C6, Canada; gtaggar7@gmail.com (G.T.); attiq.muhammad@canada.ca (M.A.R.); 2Department of Pathobiology, Ontario Veterinary College, University of Guelph, Guelph, ON N1G 2W1, Canada; pboerlin@uoguelph.ca

**Keywords:** carbapenemases, *Enterobacteriales*, human, animal, food, environment

## Abstract

The Enterobacteriales order consists of seven families including *Enterobacteriaceae*, Erwiniaceae, Pectobacteriaceae, Yersiniaceae, Hafniaceae, Morganellaceae, and Budviciaceae and 60 genera encompassing over 250 species. The *Enterobacteriaceae* is currently considered as the most taxonomically diverse among all seven recognized families. The emergence of carbapenem resistance (CR) in *Enterobacteriaceae* caused by hydrolytic enzymes called carbapenemases has become a major concern worldwide. Carbapenem-resistant *Enterobacteriaceae* (CRE) isolates have been reported not only in nosocomial and community-acquired pathogens but also in food-producing animals, companion animals, and the environment. The reported carbapenemases in *Enterobacteriaceae* from different sources belong to the Ambler class A (*bla*_KPC_), class B (*bla*_IMP_, *bla*_VIM_, *bla*_NDM_), and class D (*bla*_OXA-48_) β-lactamases. The carbapenem encoding genes are often located on plasmids or associated with various mobile genetic elements (MGEs) like transposons and integrons, which contribute significantly to their spread. These genes are most of the time associated with other antimicrobial resistance genes such as other β-lactamases, as well as aminoglycosides and fluoroquinolones resistance genes leading to multidrug resistance phenotypes. Control strategies to prevent infections due to CRE and their dissemination in human, animal and food have become necessary. Several factors involved in the emergence of CRE have been described. This review mainly focuses on the molecular epidemiology of carbapenemases in members of *Enterobacteriaceae* family from humans, animals, food and the environment.

## 1. Introduction

The actual pandemic outbreak of the COVID-19 killing several thousands of people along with its serious negative global economic impacts worldwide is a clear indication that a lot of efforts need to be deployed to fight against infectious diseases and the increased global issue of antimicrobial resistance. The World Health Organization (WHO) published a global priority list of antimicrobial resistant pathogenic bacteria including some *Enterobacteriales* for which new antibiotics are urgently needed [1]. The genera within the order *Enterobacteriales* are composed of Gram-negative bacteria of class *Gammaproteobacteria*, which encompasses many harmless symbiotic and pathogenic strains, including members of the genera *Dickeya*, *Pectobacterium*, *Brenneria*, *Erwinia* and *Pantoea* [2]. The pathogenic strains mainly *Klebsiella pneumoniae*, *Yersinia pestis, Escherichia* spp*., Salmonella enterica* serovars and *Enterobacter* spp. cause a broad range of intestinal and extra intestinal diseases in humans and animals [3]. The expended spectrum cephalosporins (ESC) and cephamycins are frequently used against infectious diseases caused by *Enterobacteriaceae*. Due to the emergence of multidrug resistance, carbapenems in addition to tigecycline and colistin are among the last line of defense against *Enterobacteriaceae*, because co-resistance to both colistin and tigecycline among the carbapenem-resistant *Enterobacteriaceae* (CRE) has been rarely reported [4,5]. Carbapenems are a powerful group of broad-spectrum antibiotics which, in many cases, are the last line of defense against multi-resistant bacterial infections. Carbapenems are classified under β-lactams antibiotics, slightly different from the penicillin by substitution of a carbon atom for a sulfur atom and addition of a double bond to the five-membered ring of the penicillin nucleus (Figure 1).

Carbapenems bind tightly to the bacterial penicillin-binding proteins (PBPs), which are vital for elongation and cross-linkage of the cell wall peptidoglycan, leading to bacterial lysis [6] (Figure 2).

Currently four of carbapenems (imipenem, meropenem, ertapenem and doripenem) are approved for clinical use in the United States of America [7] and additional three of them (biapenem, faropenem, panipenem) in Canada. During the last decade, several monitoring studies have reported the emergence of carbapenem resistant *Enterobacteriaceae* (CRE) [8]. The three major mechanisms of carbapenem resistance in these bacteria include: (i) the presence of β-lactamase enzymes called carbapenemases, and (ii) the combined effect of other β-lactamases with bacterial cell membrane permeability due to alteration or mutations in the porins and/or (iii) increased efflux pump-action (Figure 3). The modification of penicillin binding proteins (PBPs) has been reported as the forth mechanism of resistance to carbapenems in Gram-negative bacteria [9].

Carbapenemases induce resistance essentially by hydrolysis of carbapenem using active catalytic substrates either serine or zinc [10] as indicated in Table 1. The non-metallo-carbapenemase-A (Nmc-A) was first described to cause carbapenem resistance in *Enterobacteriaceae* in 1993 [11]. According to the Ambler’s molecular classification, class B metallo β-carbapenemases (MBL types), class A (*Klebsiella pneumoniae* carbapenemases (KPC) types), and class D oxacillinases (OXA types) are epidemiologically important in *Enterobacteriaceae* [12,13,14]. The genes encoding these carbapenemases can be located either on the chromosome or on mobile genetic elements (MGEs) like plasmids, integrons, and transposons [15,16]. The carbapenemases KPC and NDM (New Delhi metallo β-carbapenemases) producing bacteria have shown resistance against most of the β-lactams, fluoroquinolones, and aminoglycosides [17]. However, OXA type in particular OXA-48-like carbapenemases are less active against carbapenems and can induce a high resistance level only when associated with extended-spectrum β-lactamases (ESBLs) [18,19]. The dissemination of carbapenemases genes by MGEs among clinical isolates is a source of serious public health and food safety concerns. *Enterobacteriaceae* with acquired carbapenem resistance genes have been isolated from humans, animals, food, and the environment. The aim of this review is to discuss the molecular epidemiology of carbapenemases in *Enterobacteriaceae*.

## 2. Carbapenemases-Producing *Enterobacteriaceae* from Humans

The β-lactams antibiotics such as penicillins and cephalosporins have been wildly used against pathogenic *Enterobacteriaceae* because of their broad-spectrum activity [20]. However, this practice contributed to the emergence and spread of several types of β-lactamases including ESBLs. To overcome the resistance against extended spectrum cephalosporins (ESC) and cephamycins in *Enterobacteriaceae*, carbapenems have been introduced in human medicines. According to the recent 2020 Canadian Antimicrobial Resistance Surveillance System (CARSS) annual report, the carbapenems use in human medicine has been increased from 3.0 to 6.8 defined daily doses (DDDs) per 1000 inhabitants between 2014 and 2018 in Canada. Consequently, a concomitant nine-fold increase in the number of patients colonized by carbapenemases-producing organisms (CROs) without signs of infection has been reported in Canada [20]. Globally, the population of CRE is increasing dramatically [21,22]. There could be several factors leading to the spread of carbapenemase-producing pathogens in humans. These factors include continuous exposure to antibiotics, usage of different concentrations of antibiotics, and contamination of surgical equipment used [12,23,24].

The prevalence of carbapenem-producing *Enterobacteriaceae* (CPE) is high in humans with advanced age, primarily, due to their frequent visits to hospitals, long-term stay in health care facilities, tertiary care hospitals, and teaching hospitals [25,26,27,28,29,30]. Hospital stay represents a particularly high risk to be colonize or in developing an infection with a CRE. Recent studies found that CRE colonization among ICU patients showing new emerging mechanisms of resistance continue to rise in the United States of America [31,32]. There seems to have no different in other developed countries. For instance, Canada’s Nosocomial Infection Surveillance Program data suggested that incidences of *bla*_NDM_, *bla*_KPC_, *bla*_OXA-48_, and *bla*_VIM_ producing *K. pneumoniae* and *Enterobacter* spp. increased significantly in Canadian health care facilities since 2007 [33]. These health care facilities could be a reservoir for patient spreading CPE to multiple regions. For instance, *bla*_KPC_-producing *K. pneumonia* have been reported in hospital outbreaks in many European countries such as Greece, Italy, Spain, France and Germany [3,34,35,36,37,38,39,40]. A recent study reported cases of *bla*_NDM-1_ and *bla*_KPC-2_ -producing *K. pneumonia* among transplanted patients in Brazil [41]. Although, little is known about the spread and clinical relevance of CRE in Africa, two studies reported their prevalence in hospital and community settings among several African countries [42,43]. Issues about carbapenemases include their potential link with multidrug-resistance genes on the same MGEs. For instance, the *bla*_KPC_ gene encoding KPC enzyme to hydrolyze all β-lactams was found on plasmids carrying multiple other antimicrobial resistance determinants [44,45]. Outbreaks caused by multidrug resistant and *bla*_KPC_-positive *K. pneumoniae* opportunistic pathogenic strains have been reported in North America, Europe, Asia and South America [3,46,47,48]. A recent study from China reported a *Morganella morganii* isolate, an opportunistic pathogen, harboring *bla*_NDM-5_ gene on a self-transmissible *Inc*X3 plasmid from a stool sample of a cancer patient [49].

Another important factor responsible for worldwide dissemination of CRE is the international travelling and medical tourism. There are several reports demonstrating the role of travelling to affected developed countries in the epidemiology of CRE. Pathogenic strains of *K. pneumoniae* and *Enterobacter cloacae* containing *bla*_KPCs_ have been isolated from patients from France and Greece hospitalized in New York [50,51]. Overcrowding coupled with poor sanitation conditions including inappropriate waste management system and misuse of antibiotics could play roles in the spread of antimicrobial resistance genes in general and those for carbapenemases in particular. Furthermore, urbanization and globalization are greatly involved in spreading antimicrobial resistance pathogens all over the world. Accordingly, class B MBL, *bla*_NDM-1-_producing *K. pneumoniae, Escherichia coli, E. cloacae, Citrobacter* spp., *Proteus* spp., *and Klebsiella oxytoca* strains were originally isolated in India. These same *bla*_NDM-1_ producing *K. pneumoniae* and *E. coli* strains were subsequently isolated from Sweden and UK patients who had travelled to India recently. The patients in Sweden and UK may either have been hospitalized or underwent any medical intervention in India which could led to their infection or colonization by these pathogenic strains producing *bla*_NDM-1_ [52,53,54]. A similar scenario has been reported in Canada and the United States where *bla*_NDM-1_-producing *Enterobacteriaceae* were isolated from patients who visited and received medical cares in the Indian subcontinent [55,56]. Furthermore, a recent study reported two hyper-virulent *K*. *pneumoniae* clones of ST86 harboring plasmid mediated *bla*_KPC-2,_ isolated from a Canadian patient who visited Greece [57]. Likewise, a novel *bla*_KPC-3_ variant (*bla*_KPC-50_) was recently identified in multi-drug resistant *K*. *pneumoniae* clinical isolate conferring resistance to ceftazidime-avibactam in Switzerland that was most likely acquired in Greece [58]. Several studies have demonstrated that medical tourism is another way to introduce CREs from an endemic country to a non-endemic country [59,60,61]. For instance, a case has been reported in an Israelis hospital where four non-Israelis patients, were positive for *bla*_OXA-48_-producing *Enterobacteriaceae* [61]. The *OXA*-48 positive bacteria were absent from this hospital before these patients’ admission in the Israel, two and one of them were respectively hospitalized in Jordan and Georgia where *bla*_OXA-48_ producing *Enterobacteriaceae* were prevalent, demonstrating thus the role of medical tourism in the epidemiology of CRE [60].

## 3. Companion Animals

Carbapenems are not licensed for the treatment of infectious diseases in companion animals in most of countries. As a result, pathogenic strains of *Enterobacteriaceae* causing infections in companion animals are not usually screened for carbapenemase resistance genes in veterinary laboratories. The possible way companion animals may get infected with CRE is through direct contact with colonized hosts and contaminated environment. Eventually, companion animal may become a reservoir for CPE [62,63,64,65]. For instance, a *bla*_OXA-48_ carbapenemase-producing *K. pneumoniae* has been transmitted from human to companion animals (dogs) through contaminated hands [66]. In 2015, the transmission of an *Inc*X3 plasmid bearing *bla*_NDM-5_ in *E. coli* ST167 was detected in dogs and humans in Finland [67]. A recent study found *Inc*X3 plasmid mediated *bla*_NDM-5_ in *E. coli* ST410 in four Korean dogs implying the HGT an important mechanism of their spread among companion animals. Carbapenemase-producing *E. coli* ST131 has also been reported repeatedly in both household members and their companion animals [68,69,70]. An Australian study also reported the *Inc*HI2 type plasmid bearing *bla*_IMP-4_ in *Salmonella* Typhimurium isolates from infected cats [71]. Importantly this variant showed sequence similarity to two *bla*_IMP-8_ carrying *Inc*HI2 plasmids from *Enterobacter* spp. from humans with an indication of nosocomial spread and broader risk to humans, animals and the environment. *Enterobacteriaceae* producing carbapenemase (*bla*_NDM-1_ and *bla*_OXA-48_) have been isolated from companion animals like dogs, cats, and horses in the USA, Germany, Greece and UK [66,72,73,74,75]. Another recent study in Switzerland reported the isolation of various carbapenemase-producing *Enterobacteriales* in companion animal clinics mainly associated with poor clinical practices [76]. The isolation of CRE in companion animals brings the attention to reconsider the use of any off-label use of carbapenems in the veterinary medicines. Even carbapenems are not registered to use in companion animals, these antibiotic are used as off-label for the treatment of urinary tract infections in dogs and horses and to treat after surgical procedure infections caused by multidrug resistance *E. coli* in the UK and some other European union countries [77]. Identification of CPE in companion animals could become significant for public health due to not only host-to-host transmission but also possible gene transfers between commensals and pathogens. Due to selection pressure, treated animals (pets) may become colonized with CPE that could be transmitted to human through fecal – oral contaminations.

## 4. Carbapenemases-Producing *Enterobacteriaceae* in Other Animals

Carbapenemases-producing *Enterobacteriaceae* is not only a threat to humans; but animals may also get colonized and/or affected by CPE. The CPE could be isolated from food producing animals such as chicken, swine and cattle. A geographical distribution of CPE isolates from animal clinical samples has been recently reported [9].

## 5. Food Producing Animals

Various studies reported the presence of CPE in livestock which could constitute a food safety issue. Based on these studies, *bla*_VIM-1_ and *bla*_NDM-1_ were the most prevalent carbapenemase enzymes among *Enterobacteriaceae* in food producing animals [78,79,80,81,82]. In Germany, *bla*_VIM-1_ producing *S. enterica* and *E. coli* have been isolated from swine and chicken farms. In both cases, *bla*_VIM-1_ gene was located on an *Inc*HI2 type plasmid. These isolates showed multidrug resistance due to the presence of other resistance genes including *bla*_ACC-1_, *bla*_ADD-1_ and *str*A/B in addition to the *bla*_VIM-1_ on same *Inc*HI2 type plasmids [80,81]. Apart from *bla*_VIM-1_, three *E. coli* isolates were found to carry the *bla*_NDM-5_ gene on a *Inc*X3 transferrable plasmid, including one co-harboring the colistin resistance *mcr-1* gene on *Inc*HI2 plasmid of ST446 and the other two belonged to ST2, isolated from three dairy cows in China [83]. Moreover, a novel *Inc*X3-type plasmid harboring a *bla*_NDM_ variant (*bla*_NDM-20_,) due to three point mutations compared to *bla*_NDM-1_, was recovered recently in an *E. coli* ST1114 from swine in China thus, suggesting that food-animal could be a source of new carbapenemase genes [84].

## 6. Carbapenemases Producing *Enterobacteriaceae* in Food

CREs from food-producing animals could find their way into the food chain, leading to an alarming food safety issue. For public health concerns with respect to risk of transfer to humans via the food source, several resistance surveillance systems for retail meat are in place which include carbapenem susceptibility monitoring in many countries. Interestingly, CRO could be isolated from the food that escape resistance surveillance programs such as seafood [85]. Besides live animals, several reports demonstrating the spread of CPE in retail meat in Egypt, China and Pakistan [86,87,88]. As reported in 2016 in China, the increasing spread of carbapenem and colistin-resistant *E. coli* clone ST167 from chicken meat harboring *bla*_NDM-9_ and *mcr-1* co-located on a plasmids has raised concerns worldwide because of potential transfer of this resistance plasmids to other Gram-negative pathogens and to other countries [88]. Moreover, VIM-1 producing *S. enterica* serovar Infantis have been isolated from minced pork meats in Germany [89] and NDM-1 producing *S. enterica* serovar Indiana have been isolated from chicken carcasses in China [90]. Additionally, a recent study reported the contamination of retail meat samples from pork, chicken and beef with *E. coli* and *K. pneumonia* containing *bla*_NDM_ genes located on *Inc*X3 plasmid [91]. Between 2016 to 2018, a substantial increase of such CRE from 9.4% to 22. 2% in the retail meat samples was reported.

As stated above CRE could be found in seafood as reported as VIM-1 producing *E. coli* from retail squid, sea squirt, clams and seafood medley in Germany, China and Korea [92]. Trading of seafood from endemic countries to non-endemic countries could result in the spread of carbapenem resistance genes. For instance, the Canadian Integrated Program for Antimicrobial Resistance Surveillance (CIPARS) reported carbapenem-resistant *Enterobacter* spp. in fresh and frozen raw shrimp collected imported from Southeast Asia to Canada [93]. A recent study from Netherlands reported the isolation of *E. cloacae* ST813 bearing plasmid with *bla*_IMI-2_ and a novel Ambler class A carbapenemases *bla*_FLC-1_ from a frozen vannamei white shrimp (*Litopenaeus vannamei*) originating in India. This new *bla*_FLC-1_ carbapenemases was found to be related to previously known French imipenemase (FRI), with 82% amino acid identity to *bla*_FRI-1_ and 87% to *bla*_FRI-5_ [94]. Although rare but raw milk containing OXA-48-producing *K. pneumoniae* ST530, an epidemic clone, was reported in Lebanon [95].

A role for fresh produces in spreading antimicrobial-resistant has been suggested by several studies. In China, fresh lettuce was found to be contaminated with *E. coli* ST877 co-producing NDM-1 and KPC-2, while also carrying *fosA3* and *floR* genes on a transferrable *Inc*A/C2 type plasmid [96]. Various species of CRE were recovered from leek, radish, basil, spinach, lettuce, traditional and commercial salads in Iran and OXA-48 producing *K. pneumonia* were detected in fresh vegetables from Algeria [97]. It has also been suggested that international trade of fresh vegetables and spices could be the possible route for the spread of CRE. Coriander imported from Asia to Switzerland and many other countries was positive to *bla*_OXA-181_-producing *Klebsiella variicola* [98,99]. Consequently, there is a need for resistance surveillance programs for both carbapenemase-producing pathogenic and non-pathogenic organisms in the food chain to find the potential reservoirs of carbapenemase genes and to prevent their spread from food to humans.

## 7. Carbapenemases-Producing *Enterobacteriaceae* from the Environment

As discussed above, several studies demonstrate the spread of CPE all around the world among humans and animals. However, there are very few reports on the role of environmental contamination in the spread of CPE. The environment, surrounded by the CPE carriers, may be contaminated with these bacteria and further act as a vector for their dissemination. Dissemination of environmental CPE (eCPE) can negatively impact human health [100]. The prevalence rate of eCPE is high especially around intensive care units, acute and long-term health care facilities. Exposure of health care personnel to infected patients and cleaning methods used in the health facilities could potentially be responsible for dissemination of eCPE [101]. A study reported the presence of eCPE on the bed surface, personal table and infusion pump used by one CPE-infected patient [101]. The potential presence of eCPE on health care personnel brings attention toward the importance of adopting cleaning methods used in hospitals to disinfect the surfaces and material used by CPE carriers [102]. In hospitals, carbapenems are frequently used to treat infectious diseases caused by extended spectrum cephalosporinase (ESC) producing bacteria. Carbapenems are not entirely metabolized in the body and some residues present in human excreta can get into hospital sewage. Due to selection pressure, there is a chance that pathogens present in hospital effluent may become resistant against carbapenems. It has been reported that hospital sewage may act as a reservoir for resistance genes and a point where organisms likely acquire resistance through horizontal gene transfer events [103]. For instance, NDM-4 producing *E*. *coli* with downstream bleomycin resistance gene, *ble*_MBL_ on a plasmid type associated with complete IS *Aba*125 were isolated from hospital sewage in India, suggesting hospital wastes as major reservoir of resistance genes [104]. Likewise, antibiotic residues released into the municipal wastewater along with human excreta could contribute to selection of CPE the ground and surface water sources. A VIM-1-producing resistant *K. pneumoniae* strain has been isolated from rivers in Spain, Switzerland and Sweden [105,106,107]. The presence of CRE in wastewater is a potential concern because this environment may serve as major reservoir leading to HGT events and an increased risk of carbapenem resistance spreading into the environment. The data collected from a recent US survey of seven wastewater treatment plants reported the detection of 20% carbapenem-resistant *E. coli* isolates of sequence types associated with extra-intestinal infections in humans harboring predominantly *bla*_VIM_ and *bla*_KPC_ genes [108]. Similarly, a study from the United States documented the presence of CRE including *E. coli* and *P. mirabilis* isolates harboring *bla*_IMP-27_ gene on *Inc*Q1 plasmids in both environmental and fecal samples of swine production system [109]. The presence of *bla*_KPC-2_ gene has been reported in the United States in the metagenome from the feces of beef cattle regardless of antibiotic use in the farm [110]. Moreover, two most recent studies reported the presence and survival of carbapenem-resistant organisms harboring a plasmid-borne *bla*_OXA-23_ gene from swine manure environment from a Croatian pig farm and NDM-5 producing *E. coli* ST156 from a poultry farm in China. These suggest the possible dissemination of CRE in the food chain through animal manure fertilization [111,112]. On the other hand, irrigation water and various water matrices are also considered as major source of fresh produce contamination with resistant bacteria including CRE that can potentially be transferred to the consumer especially when consumed raw [113]. Likewise, a recent study reported carbapenem resistant *K. pneumoniae* isolates from Austrian rivers showing genetic similarities to clinical isolates from hospitals raises concerns regarding the role of surface water in the of dissemination of CRE [114]. Surprisingly, tap water could serve as reservoir of community exposure to CRE in high income countries. A recent multi-state study form United States screened drinking water samples from both public and private water systems in six states and detected 6.4% CRE harboring *bla*_OXA-48_-type carbapenemase gene [115]. Another Recent study from Egypt, investigated the occurrence of β-lactamase and CRE in the integrated agriculture–aquaculture environment and isolated several *Enterobacteriaceae* strains resistant carrying predominantly the carbapenemase resistant gene *bla*_KPC_ either alone or with the β-lactamase genes (*bla*_CTX-M-15_, *bla*_SHV_, *bla*_TEM_, and *bla*_PER-1_). This study suggests transmission of the resistance genes among *Enterobacteriaceae* strains in integrated agriculture–aquaculture system with serious public health implication [116].

## 8. Molecular Epidemiology of Carbapenem Resistance (CR) Genes

**Spread of CRE**: The most frequently identified carbapenem genes are the ambler class A including *bla*_KPC_ followed by class B metallo- beta lactamases (MLBs) such as *bla*_NDM_, and the class D OXA-type gene like *bla*_OXA-48_ (Table 1). Since its identification in the North Carolina, USA in 1996, in *K. pneumonia* patient, the epidemiology of *bla*_KPC_ producing isolates has expanded mostly globally in Africa, America, Asia, Australia, and Europe especially the clonal group (CG) 258, which includes the lineages ST258 and ST11). Since 2015, several outbreaks related to ST258 have been documented in hospital of Israel closely related to strain found in New York [117]. Likewise, the class B metallo- beta lactamases genes, including *bla*_NDM_, *bla*_SME_, *bla*_GES_, *bla*_VIM_, and *bla*_IMP_, have also been disseminated worldwide, *bla*_NDM-1_ being the most prevalent worldwide [118]. The gene *bla*_NDM-1_ was first detected in a *K. pneumonia* isolate from a Swedish patient of Indian origin in 2008 in a multidrug-resistant *K. pneumoniae*, suffering from urinary tract infection acquired in India [119]. Subsequently, the *bla*_NDM-1_ was also found in *K. pneumonia* isolate from Croatia, from a patient arrived from Bosnia and Herzegovina. The second geographical origin of *bla*_NDM-1_ considered to be eastern Balkans. In May 2010, a case of infection with *bla*_NDM-1_ producing *E. coli* was reported in Coventry in the United Kingdom. Moreover, the *bla*_NDM-1_ gene was also reported in Australia, Austria, Belgium, China, Canada, France, Germany, Hong Kong, Japan, Netherland, Norway, and Sweden. Likewise, the *bla*_SME_ were reported in England with sporadic reports of such gene in USA. Since 2004 and it has also been reported in Argentina, Australia, Brazil, Canada and Switzerland but the increasing number of this gene have been reported in America and UK [120]. The *bla*_VIM_ gene was originally described in Italian *Pseudomonas aeruginosa* in the mid-1990s and afterwards CRE carrying *bla*_VIM_ were predominantly reported in Europe and large number of cases were also reported in other countries including Africa, Taiwan, Mexico, Saudi Arabia and the United States. However, the *bla*_VIM_ gene was found in Greece with more than 48 variants of this gene showing global dissemination [121]. Furthermore, the *bla*_IMP_ gene was first originated in south pacific Asia predominantly among isolates of *E. coli* and *Enterobacter* on class 1 integrons and genomic evidences are now emerging of its dissemination globally [122]. Finally, among the various variants known of Class D today, the *bla*_OXA-48_ gene was first identified in Turkey in 2001. Although endemic during that time mostly in Turkey and Malta but since 2015, it has been spread to multiple countries including Singapore, Algeria, Korea, south Africa and have recently emerged in Canada and other western countries.

**CRE carriage:** Several studies particularly in China and also around the globe have recently attempted to investigate the carriage of CRE in animals and humans that possibly contributes to its dissemination in societies and healthcare facilities [123,124]. For instance, a recent case study in Singapore determined the duration of CRE carriage among various hospital workers by investigating 21 CRE carriers for more than 1 year. The authors of this study reported a mean carriage duration of 86 days among hospital workers and further suggested that the prolonged carriage could be associated with use of antimicrobial drugs and the probability of decolonization in a year was 98.5% [124]. Despite reports in healthcare facilities, a recent study by Zhai, et al. [125] monitored the carriage of CRE in a Chinese poultry farm for over a year (January 2017 to April 2018). During this period these authors collected 350 cloacal samples from four broiler farms in additional to 582 environmental samples and found that CRE negative 1-day-old broilers acquired *bla*_NDM_ within 24 h of transfer. Furthermore, the same study also analyzed the persistence and transmission of *bla*_NDM_-producing bacteria in a Chinese poultry farm and found that the *Inc*X3 plasmid accounts for 71% of *bla*_NDM_ carriage that persisted in farm over 16 months and about 20% also carried either the colistin resistance *mcr-1* or *mcr-8* gene. This study further suggests that the contaminated in-house environment contributes to the persistence and transmission of *bla*_NDM_ producing bacteria in Chinese poultry farms.

**Food chain transmission:** The rise of CRE in food-producing animals and food supply is of growing concern globally and, given the risks of CRE to human health, there have been a zero-tolerance policy and an international ban on the sale of food items contaminated with CRE in several countries [126]. Therefore, prevention of CRE occurrence and spread in food-producing, wildlife and companion animals is a major public health priority to protect both persons with direct exposure and consumers. Authorities in European countries have already reacted by establishing active carbapenem resistance surveillance programs targeting food-producing animals mainly poultry, pork, cattle, and retail meat products [127]. There is concern that without these programs the presence of carbapenem genes in bacteria has the potential to enter the food supply undetected and subsequently transmission to humans [19]. For example, in 2014 a study by Morrison and Rubin [85], reported a 3.94% prevalence of CPOs in seafood imported in Canada that could pose a potential risk for transfer to clinically relevant bacteria and eventually to humans which is certainly of concerns. Strong evidences exist that the transmission of resistance might have taken place from animals to humans. Although many of these evidences were not direct and were based on the similarities between the carbapenem resistance genes of bacteria isolated from food-producing animals including poultry, pigs, cattle and from humans having close contact with these animals such as farm workers, animal caretakers and their family members [128]. To support risk assessment for zoonotic CRE, a comprehensive systematic review on CRE that carried *bla*_VIM_, *bla*_KPC_, *bla*_NDM_, *blaIMP* and *bla*_OXA_, from wildlife, food-producing, and companion animals was conducted recently by Köck, el al [72]. This study was primarily focused on the dissemination of CRE in livestock, food items including seafood, companion animals and their potential of transmission to exposed humans. The most prevalent carbapenem genes were *bla*_VIM_, *bla*_KPC_, *bla*_NDM_, *bla*_OXA_, and *blaIMP* in *E. coli* and *K. pneumoniae* isolates. Interestingly, only two independent studies reported that 33–67% of exposed humans on poultry farms carried CRE closely related to isolates from the poultry farm environment suggesting fecal-oral transmission or transmission via the food sources has the higher potential to spread CREs to healthier population. Wang, et al. tested six fecal specimens from farmers and workers from a Chinese commercial chicken farm, of which three (50%) were positive for plasmid-borne *bla*_NDM-5_ producing *E. coli* and demonstrated that two of the farmers isolates shared sequence types (ST10, ST746) with isolates from local dogs, flies, and chickens and clustered closely together, suggesting exposure and transmission to human on the farm environment. In their follow up study, Wang et al. [129] demonstrated clonal commonality between *bla*_NDM_-positive *E. coli* isolates from chicken farms, slaughterhouses, supermarkets, and humans, belonging to sequence types ST10 and ST156. Recently, another study in Egypt by Elshafiee et al. [130] provided strong evidences of direct transmission of CRE to humans from farm animals. Another study by Li et al. [124], investigated the prevalence, risk factors, and drivers of CRE transmission between humans and their backyard animals in rural China that provided direct evidence of inter-host transmission of *bla*_NDM_ producing *E. coli* between humans and backyard animals. Moreover, the incidence of carbapenem resistant genes in food-producing animals has also been reported from several other countries with chickens and pig being the most investigated species, in which carbapenem resistant genes has been most frequently detected [128].

**Detection of CRE**: In the recent past, several molecular genotyping methods such as microarrays, single and multiplex polymerase chain reaction (PCR) assays have been used to detect the common carbapenemases, including *bla*_KPC_, *bla*_NDM_, *bla*_IMP_, *bla*_VIM_ and *bla*_OXA-48_, in bacterial isolates or directly from clinical specimens [131]. However, these molecular methods can accurately detect specific carbapenemase genes but cannot detect novel carbapenemase genes. On the other hand, whole genome sequencing (WGS) is a powerful new method with vastly improved resolution over current gold-standard techniques that not only identifying CRE but also providing detailed insights into their evolution and dissemination [132]. WGS potentially represent the ultimate molecular detection by probing the complete genomic content, chromosomal and extrachromosomal, of bacteria for the detection of carbapenem resistant genes. Moreover, WGS provide an opportunity to extrapolate additional information, including strain relatedness, molecular epidemiology, plasmids of replicon types harboring the carbapenemase, prediction of factors influencing carbapenem resistance for example point mutation and presence of other resistance factors and data can be analyzed in real-time or stored for future analysis. There is no doubt that WGS enhances our ability to characterize and resolve outbreaks of carbapenem resistant bacterial populations, understand and predict epidemiological trends, and create new machine learning tools for rapid detection of novel variants. As new algorithms are developing for tracking of transmission of carbapenem resistant bacteria in various fields including health care facilities and agricultural products (poultry, beef, and pork), likely to become increasing efficient and interpreting with the level of precision necessary to guide the modification of infection control procedures and food safety measures to limit the spread of CPO, allowing WGS to gain broader acceptance [133].

**Role of insertion sequences, transposons and integrons:** Dissemination of CPE among humans and animals has been enhanced by the horizontal transfer of CR genes on MGEs such as insertion sequences, transposons and integrons and their associated plasmids. [16,50,134,135,136,137]. The genetic analysis of *bla*_IMP_ and *bla*_IVIM_ from endemic areas in Southern Europe and Southeast Asia showed an association between both genes with class 1 integrons [138]. Molecular typing method such as PCR based replicon typing and plasmid multi-locus sequence typing (pMLST) methods have been used to identify a range of plasmid incompatibility groups responsible for the epidemiology of CR genes [139,140,141]. Most of the CR genes associated with the Ambler class A, class B, and Class D β-lactamases are located on plasmids, except Ambler class A CR genes (*bla*_SME-1 to 3_, *bla*_IMI_ and *bla*_NMC-A_), which are usually located on the chromosome (Table 1). Some *bla*_KPC_, *bla*_VIM_, *bla*_IMP_, and *bla*_OXA_ type carbapenemases genes have been located within transposons on plasmids which are responsible for the spread of such CR genes among *Enterobacteriaceae.* In addition, clonal expansion also acts as a way to spread the CR genes as described for *bla*_KPC_ in *K. pneumoniae* ST258 [142] and *bla*_NDM-5_ in *E. coli* on the *Inc*X3 type plasmid [143]. The insertion sequences and transposons play a role in the genetic variability. Accordingly, IS*Aba*1/3 and IS*Aba*125 have been reported to induce expression of several carbapenem resistance genes such as *bla*_OXA-23-51-66_ -like and *bla*_NDM-1_ in *A. baumannii* and *K. pneumoniae* [144,145,146,147,148]. Similarly, a novel Tn*4401* transposon variant (Tn*4401*h, a188-bp deletion) on conjugative plasmids has been associated with enhanced *bla*_KPC_ expression and the degree of phenotypic resistance to carbapenemases in *K. pneumoniae* [149].

**Plasmid:** Plasmids belonging to different incompatibility groups play an important role in the epidemiology of CR genes. Plasmid-mediated spread of *bla*_NDM-1_ gene in *Enterobacteriaceae* has attracted attention, primarily, due to its predominant presence in nosocomial isolates (*K. pneumoniae*), as well as in community-acquired isolates (*E. coli*). Another characteristic is the propensity of conjugative plasmids to transfer resistant genes among other *Enterobacteriaceae* [150,151]. The *bla*_NDM-1_ gene was first detected in New Delhi, and in the Indian subcontinents [52]. Since then it has been isolated in UK, Pakistan, Bangladesh, Central and South America, the US and Canada [151]. In this case, urbanization and international travelling were described to contribute in the propagation of the bacterial clones and plasmids with *bla*_NDM-1_ gene from Indian subcontinent to other countries and continents [151]. Genetic studies demonstrated that the *bla*_NDM-1_ gene was associated with different bacterial clones and has spread among *Enterobacteriaceae* and *non-Enterobacteriaceae* species. This *bla*_NDM-1_ gene is carried by a broad range of multidrug plasmids belonging to the *Inc*A/C, *Inc*L/M, *Inc*N and *Inc*F incompatibility groups [45]. These plasmids with *bla*_NDM-1_ gene, also harbor other genes conferring resistance to aminoglycosides, quinolones, trimethoprim, sulphonamides, tetracyclines, colistin and heavy metal [3,44,152,153]. Moreover, *bla*_NDM-1_ has been identified in *E. coli* ST90, ST131 and *E. cloacae* ST231 whereas *bla*_NDM-5_ gene in ST101, ST167, and ST405 clones bearing diverse transmitting vectors were responsible for community acquired infections [154,155]. Thus, infections caused by NDM pathogens could be difficult to be treated [52,156,157].

Some CR genes including *bla*_OXA-48_ are associated with a single type of plasmid incompatibility group in *Enterobacteriaceae* [39,158]. The dissemination of *bla*_OXA-48_ gene among *K. pneumoniae, E. coli* and *E. cloacae* is due to the presence of this gene on mono-drug resistance plasmids belonging to the *Inc*L/M incompatibility group sharing common characteristics such as self-transferability, no additional antibiotic resistance genes and size of plasmid range between 60–70 kbp [45]. In companion animals, the *bla*_OXA-48_ gene was found to be located on a multidrug-resistance plasmids belong to same *Inc*L/M incompatibility group also carrying and extended spectrum β-lactamases and Amp C β-lactamases [45,66,159].

The *bla*_KPC_ genes have usually been located on self-transferable and multidrug-resistant large size plasmids, which are varying in size as well as phage-like plasmids [160,161]. The transferability of the *bla*_KPC_ plasmids has been reported between *C. freundii* and *K. oxytoca* belonging to two different bacterial genera [162,163]. These plasmids also carry additional antibiotic resistance genes for aminoglycosides, ESBLs and fluoroquinolones [164]. Different *K. pneumoniae* clones and different strains may carry KPC β-lactamase. In all reported cases, the *bla*_KPC_ gene has been associated with the Tn*3* type transposon named Tn*4401*. This transposon may be a major source for transmission of *bla*_KPC_ gene [160].

## 9. Conclusions

The driving factor for the continuous increase in antibiotic resistance is antimicrobial usage in human and veterinary medicine as well as in agriculture. The European Surveillance of Antibiotic Consumption Network data in 2017 reported that in 10 out of 25 countries, consumption of carbapenems among humans increased since 2012, while only one country (Portugal) showed a decreasing trend during the same period (https://ecdc.europa.eu/en/publications-data/summary-latest-data-antibiotic-consumption-eu-2017). The carbapenems resistance in addition to ESC and cephamycins resistance in *Enterobacteriaceae* represent an important threat for public health. Out of the three classes, the class A carbapenemases KPC has spread globally and become more prevalent in the USA and Greece. The class B carbapenemase VIM and IMP has also been reported worldwide, but seem more endemic in Taiwan, Japan, and some European countries. The carbapenemase KPC, IMP, VIM, NDM, and OXA types have been mainly reported in nosocomial *K. pneumonia* strains. Pathogenic *E. coli* strains carrying *bla*_NDM-1_ and *bla*_OXA-48_ genes also have been found in community acquired infections. Therefore, proper identification and surveillance programs of carbapenem resistance pathogens and non-pathogenic strains have become necessary to support the control of CRE infections in both animals and humans. Source attribution studies along with developing alternative infection control strategies are warranted.

## Figures and Tables

**Figure 1 antibiotics-09-00693-f001:**
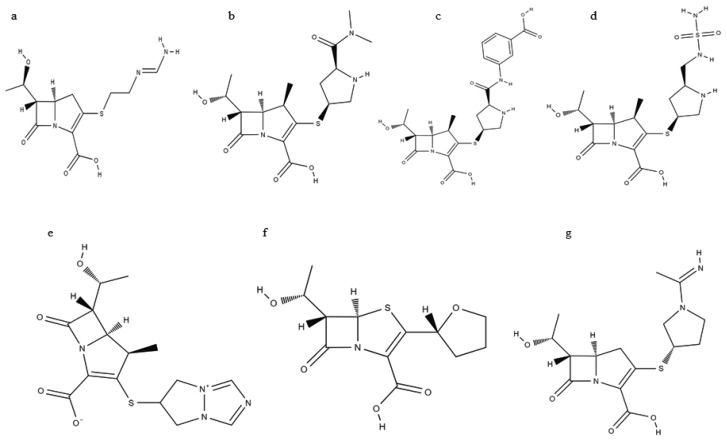
Chemical Structures of various carbapenems: (**a**) Imipenem; (**b**) Meropenem; (**c**) Ertapenem; (**d**) Doripenem; (**e**) Biapenem; (**f**) Faropenem; (**g**) Panipenem, obtained from the National Center for Biotechnology Information (NCBI) PubChem database.

**Figure 2 antibiotics-09-00693-f002:**
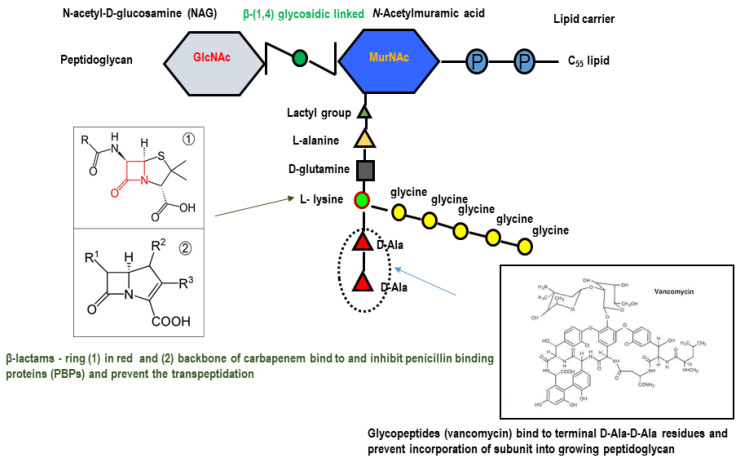
Mechanism of action of β-lactam antibiotics compared to that of vancomycin on the bacterial cell wall. Beta-lactams bind to and inhibit enzymes (PBPs: transpeptidases) which catalyse the final crosslinking (transpeptidation) of the nascent peptidoglycan layer which disrupt cell wall synthesis. Updated from Neu and Gootz, 1996, Ch. 11. Antimicrobial Chemotherapy in Medical Microbiology. 4th edition. Baron, editor. Galveston (TX): University of Texas Medical Branch at Galveston, USA.

**Figure 3 antibiotics-09-00693-f003:**
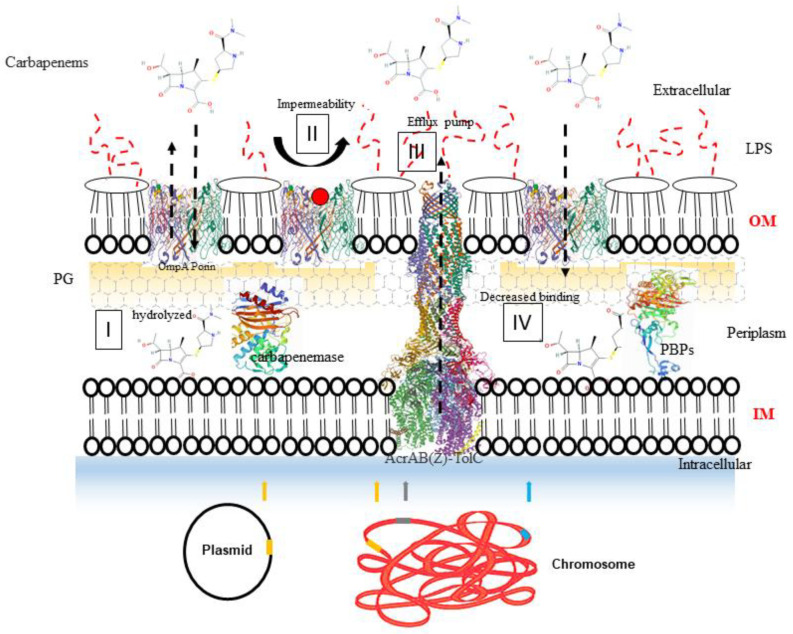
Major mechanisms associated with carbapenem-resistance in Gram-negative bacteria. (I) Production of carbapenemases enzyme from gene located on chromosome or plasmid that hydrolyze carbapenem antibiotics shown in golden rectangles. (II) Decreased permeability of the outer membrane due to structural mutations in porins (modified porins shown as red circle). (III) Drug efflux pumps. The blue and grey rectangles represent the chromosomal loci that encode various membrane associated proteins. Abbreviations: LPS, lipopolysaccharides; OM, outer membrane; IM, inner membrane; PG, peptidoglycan.

**Table 1 antibiotics-09-00693-t001:** Characteristics of the three most common classes of carbapenemases in *Enterobacteriales.*

Ambler Class.	Functional Class ^a^	Representative Gene ^b^	No. of Variants ^c^	Active Site ^d^	Substrate	Inhibitor(s)	Genetic Location	Species of Origin
A	2F	KPC	22	Serine	carbapenems, cephalosporins, Penicillins	Clavulanic acid	Chromosomally encoded; *Inc*FIIK2, *Inc*F1A, *IncI*2, multiple types; Tn*4401*	*Klebsiella pneumoniae*
		IMI	9				Chromosomally encoded, *Inc*F	*Enterobacter cloacae*
		SME	5				Chromosomally encoded, SmarGI1 novel genomic island	*Serratia marcescens*
		NMC-A	1				Chromosomally encoded	*Enterobacter cloacae*
		GES	27				Class I integrons	*Pseudomonas aeruginosa*
B	3	NDM	16	Zinc	Most β-lactams including carbapenems	EDTA	*Inc*A/C, multiple; IS*Aba125*; Tn125	*Klebsiella pneumoniae*
		IMP	56				*Inc*L/M, *Inc*A/C, multiple types; class I integrons	*Serratia marcescens*
		VIM	48				*Inc*N, *IncI*1, multiple types; class I integrons	*Pseudomonas aeruginosa*
		GIM	2				class I integrons	*Pseudomonas aeruginosa*
		SPM	1				Plasmid-mediated	*Pseudomonas aeruginosa*
		KHM	1		Most β-lactams except monobactams		Plasmid-mediated	*Citrobacter freundii*
		CcrA	1		Most β-lactams including carbapenems		Chromosomally encoded	*Bacteroides fragilis*
		BcII	1				Plasmid-mediated	*Bacillus cereus*
		CphA	8				Plasmid-mediated	*Aeromonas hydrophilia*
		L1					not determined	*Stenotrophomonas maltophilia*
D	2	OXA	489	Serine	Most β-lactams including carbapenems	Clavulanic acid	IncL/M, Tn1999, IS1999, ColE plasmids, Tn2013, IS*Ecp1*, IS*Aba125*	*Klebsiella pneumoniae*

^a^ cited in [12], ^b^ Most common carbapenemases identified, ^c^ Based on the Comprehensive Antibiotic Resistance Database (CARD v 3.0.2), ^d^ Carbapenemases induce resistance essentially by hydrolysis of carbapenem using active catalytic substrates either serine or zinc.

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
