# Peer review of "Molecular Epidemiology of Carbapenemases in Enterobacteriales from Humans, Animals, Food and the Environment"

_antibiotics, 2020, doi:10.3390/antibiotics9100693_

Round 1

Reviewer 1 Report

This review mainly focuses on the molecular epidemiology of carbapenemases in  members of Enterobacteriaceae family from humans, animals, food and the environment. Increasing resistance to carbapenem antibiotics is a significant public health problem.

Comments and Suggestions for Authors

  1. Fig. 1.- the sulfur atom in the structure of the compounds is not clearly visible, I suggest changing the colour to orange i.e.
  2. Row 312 - Pseudomonas - should be capitalized
  3. Table 1 is complex and not completely clear, includes columns characterizing carbapenemases activity, but the molecular mechanism of carbapenem hydrolysis has not been described, the information on carbapenemase active sites is very general. In my opinion, a synthetic description of the mechanisms of carbapenem hydrolysis by carbapenemases would be a good introduction to the information contained in Table 1.
  4. In the notes below table 1:
    a/ abbreviation A: Bush K, Jacoby GA. Updated functional classification of β-lactamases. Antimicrob Agents Chemother 422 2010; 54: 969–976; - add an item in the cited references [12]

b/ the description of the abbreviation D - is inappropriate with regard to the mechanism of carbapenem hydrolysis by carbapenemases

  1. Like all the above, the plasmid paragraph should be bold
  2. Row 763 - reference 91 - letters should be changed according to the instructions for authors

Author Response

REVIEWER #1 (Comments and Suggestions for Authors):

This review mainly focuses on the molecular epidemiology of carbapenemases in members of Enterobacteriaceae family from humans, animals, food and the environment. Increasing resistance to carbapenem antibiotics is a significant public health problem.

Response: We thank this reviewer for recognizing the importance of our work.

Minor issues:

  1. 1.- the sulfur atom in the structure of the compounds is not clearly visible, I suggest changing the colour to orange i.e.

Response: As suggested, In Fig.1, the sulfur atom in the structure compound has been changed to black fonts

  1. Row 312 - Pseudomonas- should be capitalized

Response:  Pseudomonas – has been capitalized (Line 339)

  1. Table 1 is complex and not completely clear, includes columns characterizing carbapenemases activity, but the molecular mechanism of carbapenem hydrolysis has not been described, the information on carbapenemase active sites is very general. In my opinion, a synthetic description of the mechanisms of carbapenem hydrolysis by carbapenemases would be a good introduction to the information contained in Table 1.

Response: As suggested by the reviewer, a description has been provided in the table 1. (Line 102-106)

  1. In the notes below table 1:
    a/ abbreviation A: Bush K, Jacoby GA. Updated functional classification of β-lactamases. Antimicrob Agents Chemother 422 2010; 54: 969–976; - add an item in the cited references [12]

Response: It has been cited in the references (Line 102)

b/ the description of the abbreviation D - is inappropriate with regard to the mechanism of carbapenem hydrolysis by carbapenemases

Response: As suggested by the reviewer, the description has been revised (Line 105-106).

  1. Like all the above, the plasmid paragraph should be bold

Response: As suggested, the plasmid paragraph is changed to bold (Line 436)

  1. Row 763 - reference 91 - letters should be changed according to the instructions for authors

Response: Reference 91 has been corrected (Line 773)

Reviewer 2 Report

The manuscript from Taggar er al is a good review related with the carbapenemases. It focuses in the epidemiology of this problem regarding all the agents implied as humans, food, environment..

The authors make an effort to compile many recent studies in the field, 177 papers are cited and many of them from recent years. Therefore, this study will be useful for the researchers and medical workers to improve the efficiency in the measures used to fight against this increasing problem in our hospitals.

The scheme of the manuscript is adequate, but I have found some minor aspects to improve it before publication.

The figures are not of enough quality to be published and should be modified as follows:

- In figure 1 the sulfur residue of all the structures should be in another color (no that yellow) because is almost invisible.

- Figure 2: the structure of vancomycin molecule is not good enough and the link ( https://en.wikipedia.org/wiki) in the legend in red is not adequate.

- Figure 3: In the plasmid and chromosome are marked three different rectangles and their corresponding arrows in several colors (yellow,  blue and grey). In the legend should be explained what represent them.

Besides in Figure 3, there are three mechanisms represented associated with the resistance but in the text only two mechanisms are mentioned. The efflux pump are absent in the text. (lines 66-69).

In lines 77-78 reads “…using active catalytic substrates either serine or zinc.” I presume that the authors mean “…using active catalytic sites either with serine or zinc” as is indicated in Table 1. And the reference to Table 1 should be in this point of the manuscript

Line 312: Pseudomonas aeruginosa is bad written

In line 315 48 variants of blaVIM are reported but in Table 1 only 23 variants are included. Please, revise this data.

Author Response

REVIEWER #2 (Comments and Suggestions for Authors):

The manuscript from Taggar er al is a good review related with the carbapenemases. It focuses in the epidemiology of this problem regarding all the agents implied as humans, food, environment.

The authors make an effort to compile many recent studies in the field, 177 papers are cited and many of them from recent years. Therefore, this study will be useful for the researchers and medical workers to improve the efficiency in the measures used to fight against this increasing problem in our hospitals.

Response: Thanks for the summary and your valuable comments/suggestions.

The scheme of the manuscript is adequate, but I have found some minor aspects to improve it before publication.

Minor issues:

The figures are not of enough quality to be published and should be modified as follows:

Response: The quality of the figures has been improved in the revised version of the manuscript

- In figure 1 the sulfur residue of all the structures should be in another color (no that yellow) because is almost invisible.

Response: In Fig.1, the sulfur residue in the structures has been changed to black font for more clarity

- Figure 2: the structure of vancomycin molecule is not good enough and the link ( https://en.wikipedia.org/wiki) in the legend in red is not adequate.

Response: As pointed out by the reviewer, In Figure 2, the structure of vancomycin molecule is appropriate and has been describe in the following book chapter: Neu H. C. and T. D. Gootz. 1996. Ch. 11. Antimicrobial Chemotherapy in Medical Microbiology. 4th edition. Baron S, editor. Galveston (TX): University of Texas Medical Branch at Galveston, USA. https://www.ncbi.nlm.nih.gov/books/NBK7986 and the link updated (https://en.wikipedia.org/wiki/Beta-lactam)

- Figure 3: In the plasmid and chromosome are marked three different rectangles and their corresponding arrows in several colors (yellow, blue and grey). In the legend should be explained what represent them.

Response: In Fig.3, description has been provided for different rectangles and their corresponding arrows in several colors (yellow, blue and grey); Line 75-76

The figure legend now read as below:

Figure 3. Major mechanisms associated with carbapenem-resistance in Gram-negative bacteria. (I)- Production of carbapenemases enzyme from gene located on chromosome or plasmid that hydrolyze carbapenem antibiotics shown in golden rectangles (II)- Decreased permeability of the outer membrane due to structural mutations in porins (modified porins shown as red circle) (III)- Drug efflux pumps. The blue and grey rectangles represent the chromosomal loci that encode various membrane associated proteins. Abbreviations: LPS; Lipopolysaccharides; OM, Outer membrane; IM, inner membrane; PG, Peptidoglycan.

Besides in Figure 3, there are three mechanisms represented associated with the resistance but in the text only two mechanisms are mentioned. The efflux pump are absent in the text. (lines 66-69).

Response: This has been included in the text “increased efflux pump-action” (Line 68) as well the possible 4th mechanism of resistance (Anderson R. E.V. and P. Boerlin. 2020. Carbapenemase-producing Enterobacteriaceae in animals and methodologies for their detection. The Canadian Journal of Veterinary Research, 84:3–17. in Gram negative bacteria)

In lines 77-78 reads “…using active catalytic substrates either serine or zinc.” I presume that the authors mean “…using active catalytic sites either with serine or zinc” as is indicated in Table 1. And the reference to Table 1 should be in this point of the manuscript

Response:  The sentence has been formatted as “using active catalytic sites either serine or zinc” as is indicated in Table 1” (Line -78-79) and the table is placed where it was cited first in the text.

Line 312: Pseudomonas aeruginosa is bad written

Response:  Pseudomonas – has been capitalized (Line 339)

In line 315 48 variants of blaVIM are reported but in Table 1 only 23 variants are included. Please, revise this data.

Response: The data has been revised in the table and now it reads as 48 (Table.1)